# FedPAM: Bridging Local Divergence in Federated Learning via Personalized Adjustment Matrix

## Abstract

Data distribution divergence across clients often leads to misalignment between global federated models and local decision boundaries. While existing personalized federated learning approaches attempt to mitigate this through feature alignment or multi-head personalization, they typically introduce additional communication and local computation overhead, which in turn limits their effectiveness under severe heterogeneity. In this work, we introduce FedPAM, a complementary perspective that addresses this challenge through a client-specific Personalized Adjustment Matrix (PAM) combined with a contrastive alignment objective, achieving robust personalization with minimal additional cost, while keeping the standard FedAvg communication protocol unchanged. Experiments across diverse benchmarks confirm that FedPAM improves upon competitive personalized FL baselines, showing pronounced advantages in highly heterogeneous conditions.

## 1 Introduction

Machine learning in cross-silo and cross-device settings is increasingly constrained by data privacy concerns, motivating federated learning (FL) as a decentralized paradigm (McMahan et al., 2017; Kairouz et al., 2021; Zhang et al., 2023a; Ye et al., 2023). FL enables collaborative model training without sharing raw data, and has found applications in healthcare , finance, and IoT systems, especially under strict data protection regulations such as GDPR (Voigt & Von dem Bussche, 2017) and CCPA (de la Torre, 2018). However, the statistical heterogeneity of client data distributions poses a fundamental challenge, frequently leading to misalignment between the global model and local decision boundaries (Zhao et al., 2018), which undermines generalization and impedes effective personalization.

To address the limitations of the global model under non-IID data, personalized federated learning (PFL) has been extensively explored (Tan et al., 2022a; Li et al., 2021c; Arivazhagan et al., 2019; Husnoo et al., 2022). Existing approaches to personalized federated learning can be broadly classified into three categories. Local fine-tuning fine-tunes the global model on each client, offering flexibility but often requiring sufficient local data (Kairouz et al., 2021). Multi-head and meta-learning methods decouple a shared backbone from client-specific classifier heads, enabling personalization at the output layer (Fallah et al., 2020; Arivazhagan et al., 2019; Husnoo et al., 2022; Chen & Chao, 2021). Feature alignment strategies instead seek to harmonize representation spaces across clients by sharing auxiliary statistics or applying regularization (Long et al., 2015; Ganin et al., 2016; Tan et al., 2022a; Wu et al., 2022). While these approaches have demonstrated effectiveness, they often incur additional communication and computational costs, and their performance degrade significantly when client heterogeneity becomes severe.

Recent studies (Luo et al., 2021) highlight that the classifier head, rather than the backbone, tends to be the primary bottleneck of federated learning under non-IID conditions, given its sensitivity to distributional heterogeneity. This motivates our design of PAM, which directly adapts the global classifier to local feature distributions to reduce such mismatches. One naive solution is to give each client an independent classifier head, but this sacrifices shared knowledge, increases communication and storage costs, and risks overfitting to limited local data. Instead, we propose a lightweight Personalized Adjustment Matrix (PAM) that adapts the shared classifier's category vectors, enabling

flexible personalization while retaining the advantages of global sharing. Combined with a supervised contrastive alignment objective, FedPAM attains classifier–feature consistency in a computationally efficient way, incurs negligible additional cost, and integrates seamlessly with existing personalized FL methods.Our contributions can be summarized as follows:

- We propose FedPAM, a personalized federated learning framework that introduces a lightweight Personalized Adjustment Matrix (PAM) together with a supervised contrastive alignment objective to mitigate classifier–feature mismatch under non-IID conditions.

- We conduct extensive experiments across diverse FL benchmarks, demonstrating that FedPAM consistently outperforms competitive personalized FL baselines, with particularly pronounced gains in highly heterogeneous scenarios, and validating the effectiveness of each component through ablation studies.

## 2 RELATED WORK

**Personalized Federated Learning.** Classical FL methods such as FedAvg (McMahan et al., 2017) aim to learn a shared global model, but this often fails under highly non-IID data. To address this, personalized FL (PFL) methods have emerged along several main directions. *Meta-learning*–based approaches (e.g., Per-FedAvg (Fallah et al., 2020)) learn an initialization that quickly adapts to each client. *Regularization-based* methods (e.g., pFedMe (T Dinh et al., 2020), Ditto (Li et al., 2021b)) constrain local updates toward global knowledge while enabling client-specific solutions. *Personalized-head* designs decouple a shared backbone from client-specific classifiers, as in FedPer and FedRep (Arivazhagan et al., 2019; Husnoo et al., 2022). Other works personalize the *aggregation process*, deriving client-dependent weights or even maintaining per-client server models (e.g., FedAMP, FedFomo (Huang et al., 2021; Zhang et al., 2020)). To bridge global and local objectives, hybrid approaches like FedRoD (Chen & Chao, 2021) jointly learn shared and personalized components. Overall, most existing methods operate at the level of parameters, backbones, or aggregation. However, they often overlook the misalignment between learned features and the classifier decision space. Our method tackles this gap by introducing a lightweight adapter at the classifier–feature interface, reshaping local decision boundaries while still benefiting from global sharing.

**Feature Alignment and Adaptation.** Beyond approaches that personalize model parameters or architectural components, another line of work focuses on aligning feature spaces across heterogeneous clients. Early approaches draw on distribution alignment, for example, matching statistics with MMD or adversarial discriminators (Long et al., 2015; Ganin et al., 2016). Prototype-based methods further anchor semantics by sharing or distilling class representatives (Dai et al., 2023; Tan et al., 2022a), while contrastive formulations enhance separation between classes and have been adapted to the federated setting (Li et al., 2021a; Wu et al., 2022; Tan et al., 2022b). Normalization or adapter based techniques offer lightweight personalization of shared backbones (Li et al., 2021c; Zhang et al., 2023b). Despite their success, these approaches primarily target representation alignment; they often overlook the mismatch at *classifier feature interface*, where inconsistent decision boundaries across clients remain a key challenge. Our method builds on the spirit of contrastive learning, but goes further by directly adapting the classifier vectors, thereby reconciling both representation and classifier alignment with minimal overhead.

## 3 METHODOLOGY

**Problem Setup.** We consider a federated learning setting with $K$ clients, each holding a private dataset $\mathcal{D}_k = \{(\mathbf{x}_i, y_i)\}$ sampled from a local distribution $\mathcal{P}_k$ that may differ across clients. The global model consists of a feature extractor $f_\theta : \mathcal{X} \to \mathbb{R}^d$ with parameters $\theta$, followed by a linear classifier $W \in \mathbb{R}^{C \times d}$, where $C$ is the number of classes. Given an input $\mathbf{x}$, the feature representation is $z = f_\theta(\mathbf{x})$, and the corresponding global logits are $Wz$. We adopt the standard FedAvg communication protocol: in each round $t$, the server broadcasts the global parameters $(\theta^{(t)}, W^{(t)})$ to a sampled subset of clients. Each client updates the model locally for $E$ epochs and returns its parameters $(\theta_k^{(t+1)}, W_k^{(t+1)})$ to the server, which performs weighted averaging based on local data sizes to obtain the new global model.

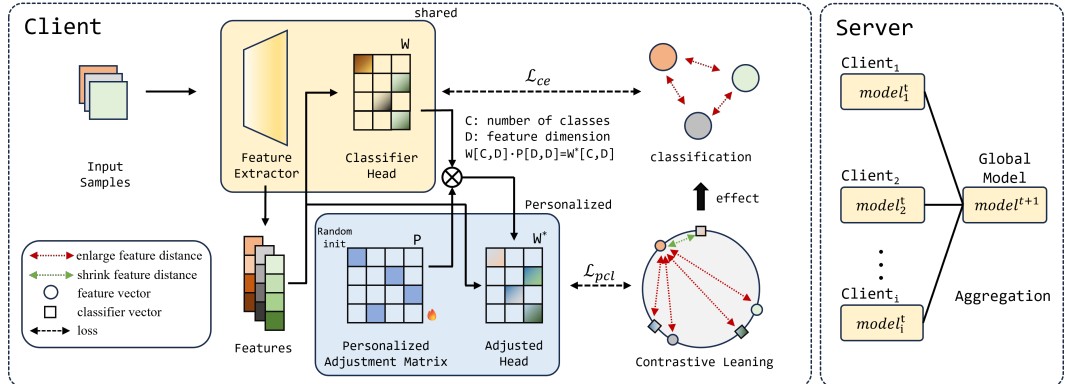

Figure 1: Overview of FedPAM. Yellow indicates shared modules, namely the Feature Extractor and the Classifier Head $W$, which are synchronized across clients. Blue indicates personalized modules, namely the PAM $P$ and the adjusted head that yields $W^*$, which remain on the client. Training employs two complementary objectives: $\mathcal{L}_{\text{ce}}$ learns a global classification objective on the shared head to align clients, whereas $\mathcal{L}_{\text{pcl}}$ learns a personalized objective on $W^*$ to fit each client's distribution. Only shared updates are uploaded for server aggregation to produce $model^{t+1}$, while personalized parameters stay on the client.

**Overview.** FedPAM provides classifier-side personalization for federated learning, as shown in Figure 1. Each client learns a lightweight adjustment matrix $P_k$ that transforms the shared classifier $W$ into an adjusted head $WP_k$ aligned with its local feature distribution. Training uses two coupled objectives: cross-entropy on the global logits $Wz$ to preserve a common decision space and a personalized contrastive loss that pulls features toward their adjusted class anchors and pushes them away from other classes. The protocol follows FedAvg without extra communication cost: the server broadcasts $\theta$ and $W$, clients optimize $\theta$, $W$, and $P_k$, and only the shared parameters return to the server for aggregation, while $P_k$ stays private. At inference, each client combines $f_\theta$ with its adjusted head to obtain personalized predictions, improving stability under heterogeneous data and partial participation while scaling to large client populations with minimal overhead.

### 3.1 FedPAM: Personalized Adjustment Matrix

**Personalized Adjustment Matrix.** In federated learning, the globally aggregated classifier $W$ often fails to align with the feature representation of each client. This mismatch arises because client data distributions can differ drastically: some classes are underrepresented on certain devices, while others exhibit distinct feature orientations or covariances compared to the global average. As a result, directly reusing the same $W$ across all clients creates misaligned decision boundaries and significantly reduces local accuracy. Importantly, we find that this bottleneck lies not in the backbone feature extractor but in the classifier head, which is especially sensitive to distribution shifts.

To address this issue, FedPAM introduces a lightweight, client-specific adjustment matrix $P_k$. Its role is to act as a local calibrator: instead of discarding the global classifier, each client gently reshapes it by computing

$$\mathbf{W}_k^{adj} = \mathbf{W}\mathbf{P}_k. \tag{3.1}$$

Intuitively, the global classifier $W$ encodes broadly transferable knowledge, while $P_k$ performs small alignment corrections, such as rotations or rescalings that make the class vectors better aligned with the client's own feature space. Since $P_k$ is trained and kept locally without being uploaded or aggregated, FedPAM preserves the exact same communication cost as FedAvg.

**Contrastive Alignment Objective.** However, adapting the classifier alone is insufficient if the local features do not naturally conform to these adjusted class directions. To further enforce compatibility, FedPAM incorporates a supervised contrastive alignment objective. This objective encourages each feature embedding to be close to its corresponding adjusted class vector while being

separated from other classes and in-batch negatives. In this way, both sides of the classification interface are aligned: $P_k$ ensures the decision boundaries match local feature distribution, while the contrastive loss pulls the features themselves toward these boundaries. At the same time, we still compute cross-entropy loss on the unadjusted global logits $Wz$, ensuring that all clients continue to optimize a common global decision space for stable aggregation.

With $P_k$ in place, the rows of $W_k^{\text{adj}}$ become client-specific *class anchors* that reflect the calibrated decision directions actually needed on client $k$. Aligning features to these anchors makes the calibration effective. For a minibatch $\mathcal{B}_k = \{(\mathbf{x}_i, y_i)\}$, let $z_i = f_\theta(x_i)$, and use $\ell_2$-normalized features and anchors,

$$\tilde{z}_i = \frac{z_i}{\|z_i\|}, \quad \tilde{w}_{k,c} = \frac{(W_k^{\text{adj}})_c}{\|(W_k^{\text{adj}})_c\|}, \quad s(a,b) = a^\top b. \tag{3.2}$$

For sample $i$, the positive is its adjusted class anchor $\tilde{\mathbf{w}}_{k,y_i}$; negatives are the other adjusted anchors $\{\tilde{\mathbf{w}}_{k,c}\}_{c \neq y_i}$ together with in-batch features of different labels. We define the *Personalized Contrastive loss (PCL)* with temperature $\tau > 0$ as

$$\mathcal{L}_{\text{pcl}} = \frac{1}{|\mathcal{B}_k|} \sum_{i \in \mathcal{B}_k} -\log \frac{\exp\big(s(\tilde{z}_i, \tilde{w}_{k,y_i})/\tau\big)}{\sum_{c=1}^C \exp\big(s(\tilde{z}_i, \tilde{w}_{k,c})/\tau\big) + \sum_{\substack{j \in \mathcal{B}_k \\ y_j \neq y_i}} \exp\big(s(\tilde{z}_i, \tilde{z}_j)/\tau\big)}. \tag{3.3}$$

This pulls features toward their calibrated class direction and pushes them away from competing classes and mismatched instances, closing the gap from the feature side while $P_k$ closes it from the classifier side. The client's local objective combines this alignment with a cross-entropy on *unadjusted* logits to preserve stable aggregation:

$$\mathcal{L}_k(\theta, W, P_k) = \frac{1}{|\mathcal{B}_k|} \sum_{i \in \mathcal{B}_k} \mathcal{L}_{\text{ce}}\big(Wz_i, y_i\big) + \lambda\, \mathcal{L}_{\text{pcl}}, \tag{3.4}$$

where $\lambda \geq 0$ balances personalization and global consistency. Client $k$ updates $(\theta, W, P_k)$ with this loss but uploads only $(\theta, W)$ for averaging; $P_k$ remains on device and is reused across rounds.

### 3.2 CONVERGENCE ANALYSIS

We provide convergence guarantees of FedPAM on the augmented local objective

$$\tilde{F}_k := \mathcal{L}_{\text{ce}}^k + \lambda \mathcal{L}_{\text{pcl}}^k,$$

where $\mathcal{L}_{\text{ce}}^k$ is the local cross-entropy objective and $\mathcal{L}_{\text{pcl}}^k$ is the PAM-based alignment regularizer. We first present a set of assumptions required to perform the convergence analysis.

**Assumption 1** (Smoothness). The loss functions $\mathcal{L}_{\text{ce}}^k$ and $\mathcal{L}_{\text{pcl}}^k$ are $L$-smooth, meaning that their gradients are Lipschitz continuous. For all clients $k$ and any $u, v \in \mathbb{R}^d$, we have

$$\|\nabla\mathcal{L}_{\text{ce}}(u) - \nabla\mathcal{L}_{\text{ce}}(v)\| \leq L\, \|u - v\|, \qquad \|\nabla\mathcal{L}_{\text{pcl}}(u) - \nabla\mathcal{L}_{\text{pcl}}(v)\| \leq L_P\, \|u - v\|.$$

Consequently, each augmented objective $\tilde{F}_k$ is $\tilde{L}$-smooth with

$$\|\nabla\tilde{F}_k(u) - \nabla\tilde{F}_k(v)\| \leq \tilde{L}\, \|u - v\|, \qquad \tilde{L} := L + \lambda L_P.$$

**Assumption 2** (Unbiased stochastic gradients). The gradient of each client is unbiased. The minibatch gradient serves as an unbiased estimator of the true gradient, meaning its expectation coincides with the exact gradient, and thus points in the correct descent direction on average. Let $g_{k,\tau}$ denote the mini-batch gradient of $\tilde{F}_k$ at local step $\tau$ from $\theta_{k,\tau}$. This is expressed as

$$\mathbb{E}[g_{k,\tau} \mid \theta_{k,\tau}] = \nabla\tilde{F}_k(\theta_{k,\tau}).$$

**Assumption 3** (Bounded Variance). The stochastic gradient has bounded variance. This means that although each mini-batch gradient is a noisy estimator of the true gradient, its fluctuations around the mean remain within a finite range. There exists a constant $\tilde{\sigma} > 0$ such that

$$\mathbb{E}\left\|g_{k,\tau} - \nabla \tilde{F}_k(\theta_{k,\tau})\right\|^2 \leq \tilde{\sigma}^2.$$

**Theorem 1** (Convergence rate of FedPAM). Under Assumptions 1–3, this theorem establishes the global convergence behavior of FedPAM. It quantifies how the average squared gradient norm decays with the number of communication rounds $T$, and thus provides a measure of how fast FedPAM approaches a stationary point. The bound highlights two key components: (i) a $\mathcal{O}(1/T)$ optimization error term, and (ii) a variance term caused by stochastic gradients. Formally, let $0 < \gamma < 2/\tilde{L}$ and define

$$\frac{1}{T} \sum_{t=0}^{T-1} \mathbb{E}\|\nabla \tilde{F}(\theta_t)\|^2 \leq \frac{\tilde{F}(\theta_0) - \tilde{F}_{\inf}}{\tilde{\beta}T} + \frac{\tilde{L}\gamma^2}{2\tilde{\beta}} E\,\tilde{\sigma}^2 + \frac{\tilde{L}^2\gamma^3}{12\tilde{\beta}} E(E-1)(2E-1)\,\tilde{\sigma}^2.$$

where $\tilde{F}(\theta) = \sum_k p_k \tilde{F}_k(\theta)$. Here $\tilde{\beta} := \gamma\left(1 - \frac{\tilde{L}\gamma}{2}\right)$, $\gamma$ denotes the learning rate of local gradient descent, $\tilde{L}$ denotes the smoothness constant from Assumption 1, $E$ is the number of local steps, and $\tilde{\sigma}^2$ is the variance bound from Assumption 3. Therefore, FedPAM achieves the $\mathcal{O}(1/T)$ convergence rate up to a constant variance term.

**Proof Sketch.** The proof follows standard FedAvg analysis: bounding descent of $\tilde{F}$ per iteration, controlling stochastic variance, and treating $P_k$ as client-local parameters that only affect $P$. Full derivations are deferred to Appendix B.

## 4 EMPIRICAL RESULTS

**Datasets and models.** We evaluate our approach in a federated learning scenario on five standard image classification benchmarks: MNIST (LeCun et al., 1998), CIFAR-10 (Krizhevsky, 2009), CIFAR-100 (Krizhevsky, 2009), CINIC-10 (Darlow et al., 2018), and Tiny-ImageNet (Le & Yang, 2015). For model architectures, we use a lightweight CNN (Krizhevsky et al., 2012) for MNIST, and adopt ResNet-18 (He et al., 2016) for the other datasets. In all cases, the backbone network is followed by a linear classifier.

**Partition protocols.** To emulate heterogeneous clients, we primarily adopt a Dirichlet-based partitioning strategy, where each dataset is split across $N$ clients according to a Dirichlet distribution with concentration parameter $\beta$ (larger $\beta$ indicates stronger non-IID). We use the *default* configuration of $\beta$=0.1 with $N$=20 clients, and also investigate a mild-skew setting with $\beta$=0.01. In addition to Dirichlet-based splits, we also consider an independent *pathological* partitioning protocol, widely used in prior federated learning literature. In this setting, each client is restricted to samples from only a small number of classes (two in our experiments), leading to highly skewed and disjoint local datasets. This setup provides a complementary perspective on extreme data heterogeneity compared to the Dirichlet-based protocol. We further vary the federation size $N \in \{10, 30, 50, 100, 200\}$ and simulate random client dropout with participation ratio $p \in [0.1, 1]$.

**Training protocol.** Each round, the server samples a fraction $r$ of clients, broadcasts the current global model, and aggregates client updates via data-size–weighted averaging; each selected client trains locally for $E$ epochs. We train for $T$=200 rounds using SGD with momentum, batch size 10, initial learning rate 0.005, and report mean test accuracy over three random seeds. Architectures and budgets are kept fixed across methods to isolate the algorithmic effect of personalization.

**Baselines and variants.** We compare against classical FedAvg/FedProx and pFL methods (Per-FedAvg (Fallah et al., 2020), pFedMe (T Dinh et al., 2020), FedAMP (Huang et al., 2021), Ditto (Li et al., 2021b), FedPer (Arivazhagan et al., 2019), FedRep (Husnoo et al., 2022), FedRoD (Chen & Chao, 2021), FedFomo (Zhang et al., 2020), FedPHP (Li et al., 2021d), pFedFDA (Mclaughlin & Su, 2024), FedAS (Yang et al., 2024), FedCP (Zhang et al., 2023b)).

## 4.1 Main Results

Across both the *pathological* and *default* partitions, FedPAM consistently achieves the best or tied-best accuracy on all datasets (Table 1). Under the *pathological* setting, it matches the strongest baseline on MNIST and brings noticeable gains on CIFAR-10 and CIFAR-100. Under the *default* Dirichlet split ($\beta$=0.1), FedPAM likewise ranks first across the board, with especially clear improvements on the more challenging datasets such as CIFAR-100, CINIC-10, and Tiny-ImageNet. On easier benchmarks like MNIST and CIFAR-10, the margins become very small, which reflects that most personalized FL methods already perform strongly in these near-saturation regimes.

**Comparison Overall.** Personalization helps substantially under heterogeneous data: Traditional FL methods such as FedAvg and FedProx lag far behind most personalized methods, especially on CIFAR-100/Tiny-ImageNet and under the pathological split. Among personalized baselines, feature-alignment approaches (e.g., pFedFDA, FedRoD) improve over early methods by encouraging consistent representation spaces, but their gains plateau on complex datasets and sometimes trail behind representation-decoupling strategies (e.g., FedRep, FedAS). FedPAM consistently outperforms both groups, indicating that its adaptive personalization better resolves client drift without being restricted to a fixed shared feature space. Performance gains are most pronounced on harder benchmarks (CINIC-10, CIFAR-100, Tiny-ImageNet), where heterogeneity and task complexity stress optimization and generalization; On easier benchmarks, improvements are limited but reliable, suggesting a near-saturation regime rather than unstable training.

Table 1: The accuracy (%) of the image classification tasks in the main experiments.

| Setting | Pathological | | | Default ($\beta = 0.1$) | | | | |
|---|---|---|---|---|---|---|---|---|
| Dataset | MNIST | CIFAR-10 | CIFAR-100 | MNIST | CIFAR-10 | CINIC-10 | CIFAR-100 | TINY |
| FedAvg | 97.93 | 55.09 | 25.98 | 98.81 | 59.16 | 32.20 | 31.89 | 19.45 |
| FedProx | 98.01 | 55.06 | 25.94 | 98.82 | 59.21 | 30.28 | 31.99 | 19.27 |
| Per-FedAvg | 99.63 | 89.63 | 56.80 | 98.90 | 87.74 | 82.31 | 44.28 | 21.81 |
| pFedMe | 99.75 | 90.11 | 58.20 | 99.52 | 88.09 | 86.59 | 47.34 | 33.44 |
| FedAMP | 99.76 | 90.79 | 64.34 | 99.47 | 88.70 | 86.21 | 47.69 | 29.11 |
| Ditto | 99.81 | 92.39 | 67.23 | 99.64 | 90.59 | 85.11 | 52.87 | 32.15 |
| FedPer | 99.70 | 91.15 | 63.53 | 99.47 | 89.22 | 83.24 | 49.63 | 33.84 |
| FedRep | 99.77 | 91.93 | 67.56 | 99.48 | 90.40 | 83.69 | 52.39 | 37.27 |
| FedRoD | 99.90 | 91.98 | 62.30 | 99.66 | 89.93 | 83.53 | 50.94 | 36.43 |
| FedFomo | 99.83 | 91.85 | 62.49 | 99.33 | 88.06 | 82.69 | 45.39 | 26.33 |
| FedPHP | 99.73 | 90.01 | 63.09 | 99.58 | 88.92 | 86.61 | 50.52 | 35.69 |
| FedAS | 99.84 | 90.01 | 67.82 | 99.65 | 90.40 | 86.71 | 60.21 | 45.87 |
| pFedFDA | 99.87 | 88.65 | 65.26 | 99.60 | 89.68 | 86.61 | 50.52 | 34.58 |
| FedCP | 99.91 | 92.67 | 71.80 | 99.71 | 90.40 | 85.16 | 57.86 | 44.18 |
| **FedPAM** | **99.91** | **93.15** | **73.24** | **99.81** | **90.60** | **88.17** | **62.73** | **47.03** |

**Effect of heterogeneity.** Table 2 shows that FedPAM consistently ranks top-1 across Dirichlet partitions of Tiny-ImageNet with varying heterogeneity. Under mild heterogeneity, several personalized approaches such as FedAS, pFedFDA, and FedCP achieve relatively strong performance, yet FedPAM still maintains a clear lead. As heterogeneity intensifies, many baselines degrade sharply; methods like FedAMP, Per-FedAvg, and FedRep in particular exhibit substantial performance drops, whereas FedPAM degrades much more gracefully. Feature-alignment strategies such as FedRoD and pFedFDA display partial robustness but still fall short of FedPAM's stability. These comparisons suggest that FedPAM is not only competitive under easier, more balanced partitions but also significantly more robust when the data distribution becomes highly skewed, preserving leading accuracy where other methods collapse.

**Scalability with client numbers.** We distinguish between Table 2 and Table 4. The former varies the *total number of clients* in the federation ($N$=10, 30, 50, 100, 200), while the latter fixes the total client population at 50 but varies the *number of clients sampled per round*, for example $N = 10|50$. These settings provide complementary views: scalability of the federated system versus robustness under partial participation.

Table 2: The accuracy (%) of the image classification tasks for heterogeneity and scalability.

| Setting Dataset | Heterogeneity TINY | | Scalability CIFAR-100 | | | | |
|---|---|---|---|---|---|---|---|
| | $\beta = 0.01$ | $\beta = 0.5$ | $N = 10$ | $N = 30$ | $N = 50$ | $N = 100$ | $N = 200$ |
| FedAvg | 15.70 | 21.14 | 31.47 | 31.15 | 31.90 | 31.95 | 31.20 |
| FedProx | 15.66 | 21.22 | 31.24 | 31.21 | 31.94 | 31.97 | 31.22 |
| Per-FedAvg | 39.67 | 16.49 | 36.12 | 43.21 | 42.51 | 34.59 | 35.31 |
| pFedMe | 41.43 | 17.43 | 44.16 | 47.11 | 48.32 | 46.41 | 39.53 |
| FedAMP | 48.42 | 12.48 | 49.31 | 45.37 | 44.74 | 40.50 | 35.43 |
| Ditto | 50.62 | 18.98 | 52.32 | 52.53 | 54.22 | 52.89 | 35.18 |
| FedPer | 51.83 | 17.31 | 50.31 | 44.98 | 44.22 | 40.37 | 34.99 |
| FedRep | 55.43 | 16.74 | 52.89 | 50.24 | 47.41 | 44.61 | 36.79 |
| FedRoD | 49.17 | 23.23 | 49.83 | 50.11 | 49.38 | 46.65 | 43.53 |
| FedFomo | 46.36 | 11.59 | 46.71 | 43.20 | 42.56 | 38.91 | 34.79 |
| FedPHP | 48.63 | 21.09 | 49.32 | 49.28 | 52.44 | 49.70 | 34.48 |
| pFedFDA | 58.52 | 27.11 | 52.04 | 50.85 | 53.71 | 38.72 | 38.08 |
| FedAS | 63.31 | 22.94 | 57.60 | 49.04 | 42.73 | 46.36 | 32.79 |
| FedCP | 56.31 | 27.66 | 58.36 | 56.93 | 55.43 | 53.81 | 44.86 |
| **FedPAM** | **63.62** | **29.80** | **61.47** | **61.31** | **58.91** | **54.63** | **46.63** |

As $N$ increases, the task becomes harder since each client has fewer samples and participates less often. FedAvg and FedProx stagnate around 31%. FedAS degrades sharply, from 57.60% at $N = 10$ to 32.79% at $N = 200$. Feature alignment methods such as pFedFDA and FedRoD perform well at moderate scales but lack stability, for example pFedFDA drops from 53.71% at $N = 50$ to 38.72% at $N = 100$. In contrast, FedPAM consistently achieves the best results, outperforming strong baselines at small scales and degrading gracefully at large scales while maintaining a clear margin. Results in Table 4 confirm this trend, FedPAM preserves top performance across $N = 10|50, 30|50, 50$, while other methods fluctuate or collapse. Partial participation can improve accuracy over full participation for $N = 10$ and $N = 50$, because stochastic regularization mitigates client drift. Overall, FedPAM scales robustly and sustains personalization quality even in large, data scarce federations.

Table 3: Test accuracy (%) on CIFAR-100 ($\beta = 0.1, N = 20$) under random dropout settings.

| Method | Dropout Ratio ($p$) | | |
|---|---|---|---|
| | $p = 1$ | $p \in [0.5, 1]$ | $p \in [0.1, 1]$ |
| Per-FedAvg | 44.31 | 43.66 | 43.63 |
| pFedMe | 48.36 | 43.28 | 41.71 |
| FedAMP | 44.39 | 42.91 | 42.92 |
| Ditto | 50.59 | 49.78 | 48.33 |
| FedPer | 44.22 | 44.12 | 44.07 |
| FedRep | 47.41 | 46.93 | 46.61 |
| FedRoD | 49.38 | 49.07 | 47.80 |
| FedFomo | 42.56 | 40.96 | 40.93 |
| FedPHP | 50.23 | 45.19 | 44.43 |
| pFedFDA | 53.71 | 52.30 | 54.22 |
| FedAS | 42.73 | 44.86 | 49.34 |
| FedCP | 55.43 | 54.68 | 54.20 |
| **FedPAM** | **58.91** | **58.11** | **57.77** |

Table 4: Test accuracy (%) on CIFAR-100 ($\beta = 0.1$) under partial participation.

| Method | Number of Clients ($N$) | | |
|---|---|---|---|
| | $N = 10|50$ | $N = 30|50$ | $N = 50$ |
| Per-FedAvg | 40.20 | 42.96 | 44.31 |
| pFedMe | 40.27 | 42.19 | 48.36 |
| FedAMP | 43.57 | 43.18 | 44.39 |
| Ditto | 48.23 | 50.98 | 54.22 |
| FedPer | 43.64 | 43.54 | 44.22 |
| FedRep | 46.85 | 47.63 | 47.41 |
| FedRoD | 46.32 | 49.15 | 49.38 |
| FedFomo | 41.53 | 40.69 | 42.56 |
| FedPHP | 45.71 | 48.65 | 52.44 |
| pFedFDA | 52.77 | 53.63 | 53.71 |
| FedAS | 58.29 | 42.23 | 42.73 |
| FedCP | 50.93 | 54.31 | 55.43 |
| **FedPAM** | **61.39** | **59.27** | **58.91** |

**Robustness to client dropout.** In real world mobile scenarios, network instability can cause clients to miss a round and rejoin later. We simulate random client dropout with probability $p$ and report results in Table 3. Accuracy typically declines as participation shrinks because aggregation uses less diverse data and client drift grows. Among baselines, FedCP remains competitive but still trails, pFedFDA and FedRoD show partial resilience yet fall short on absolute accuracy, pFedMe and Ditto degrade noticeably as $p$ increases, and FedAS fluctuates, revealing instability. In contrast, FedPAM maintains the highest accuracy with minimal variation across dropout levels. For example, it reaches 58.91% at $p = 1$ and remains 57.77% when $p$ ranges from 0.1 to 1. This stability arises

from classifier side calibration and contrastive alignment, which align features with adjusted class anchors while preserving a common decision space. Overall, FedPAM handles partial and stochastic participation reliably and sustains personalization quality under changing client availability.

## 4.2 ABLATION STUDIES

We conduct a series of ablation studies to dissect the contributions of different components in Fed-PAM and to better understand the factors influencing its performance. Specifically, we investigate (i) the individual and joint effects of the proposed PAM module and the contrastive learning objective, (ii) the sensitivity of FedPAM to the PCL weight $\lambda$, (iii) the impact of the PAM bottleneck capacity $m$, and (iv) the empirical runtime and communication cost compared with representative baselines. These analyses provide deeper insights into how each design choice affects both accuracy and efficiency, and demonstrate that FedPAM strikes a favorable balance between personalization quality and system overhead.

Table 5: Core component ablations under the default setting on CIFAR-100.

| Variant | PAM | PCL | Acc. (%) |
|---------|-----|-----|----------|
| FedAvg | × | × | 31.89 |
| PCL only | × | ✓ | 56.52 |
| PAM only | ✓ | × | 49.39 |
| **FedPAM** | ✓ | ✓ | 62.73 |

**Effect of PAM and PCL.** Table 5 presents core component ablations on CIFAR100 under the default partition with $\beta = 0.1$. PAM recalibrates the classifier, and PCL aligns features to the resulting adjusted head. Each component alone improves accuracy over FedAvg but addresses a different side of the mismatch: without PAM the class directions remain misaligned, and without PCL the features fail to conform to the calibrated directions. Used together, FedPAM attains the best accuracy, showing that classifier recalibration and post adjustment feature alignment are both necessary and mutually reinforcing.

Table 6: Sensitivity to alignment weight $\lambda$ on default setting using CIFAR-100.

| $\lambda$ | 1 | 5 | 10 | 15 | 20 | 25 | 30 | 35 |
|-----------|---|---|----|----|----|----|----|----|
| Acc. (%) | 54.03 | 56.78 | 58.67 | 60.10 | 61.08 | 61.40 | 62.73 | 62.04 |

**Effect of the PCL weight $\lambda$.** Table 6 shows the sensitivity of FedPAM to the contrastive alignment weight $\lambda$ on CIFAR-100. Accuracy rises steadily as $\lambda$ increases from 1 to 30, reaching 62.73%, then drops slightly to 62.04% at 35. This trend reflects the scale gap between the contrastive loss and the classification loss: due to normalization in contrastive objectives such as cosine similarity, their magnitude is relatively small, so a larger $\lambda$ is needed to balance the losses. Weights around 20 to 30 give the best performance, while excessively large values may cause over-regularization and reduce client-specific adaptation. Overall, FedPAM remains stable across a broad range of $\lambda$, with 30 providing the most favorable trade-off.

Table 7: Effect of PAM capacity on default setting using CIFAR-100.

| | Identity | $m$=64 | $m$=128 | $m$=256 | $m$=512 | $m$=1024 |
|---|----------|--------|---------|---------|---------|----------|
| Acc. (%) | 56.52 | 60.20 | 60.54 | 61.01 | 62.73 | 61.44 |
| $\Delta$Acc | – | ↑3.68 | ↑4.02 | ↑4.49 | ↑6.21 | ↑4.92 |

**Ablation on PAM capacity.** Table 7 examines the effect of the PAM bottleneck size $m$ on CI-FAR100. Compared with the identity mapping, increasing capacity steadily improves accuracy from

56.52% to 62.73% at $m = 512$. The gains are substantial at small to medium sizes, for example $\Delta$Acc of 3.68% at $m = 64$ and 4.49% at $m = 256$, and they peak at 6.21% at $m = 512$. Further enlarging to $m = 1024$ yields a slight drop to 61.44%, suggesting diminishing returns and mild overfitting or optimization inefficiency at very high capacity. In practice $m$ is typically chosen to match the feature extractor output dimension $d$; when a different $m$ is desired, an MLP after the feature extractor can project features from $d$ to $m$ so that PAM capacity changes without altering the shared backbone. Overall, a moderate to large PAM capacity is beneficial, with $m = 512$ providing the best trade off between expressiveness and stability.

Table 8: Empirical local training time and communication cost on CIFAR-100. Here $\Sigma$ denotes transmitting the full model parameters, $\alpha\Sigma$ ($0 < \alpha < 1$) denotes a fraction of parameters such as a personalization head, $S$ denotes an additional state parameter, $d$ denotes the feature dimension of the backbone output, and $O(\cdot)$ indicates the asymptotic size of extra transmitted parameters.

| Method | Comm. Cost | Local Time | Method | Comm. Cost | Local Time |
|---|---|---|---|---|---|
| FedAvg | $\Sigma$ | 96s | FedProx | $\Sigma$ | 119s |
| Per-FedAvg | $\Sigma$ | 115s | FedPer | $\alpha\Sigma$ | 246s |
| FedRep | $\alpha\Sigma$ | 246s | FedRoD | $\Sigma$ | 104s |
| FedPHP | $\Sigma$ | 243s | pFedFDA | $\Sigma + O(d^2)$ | – |
| FedAS | $\Sigma + S$ | 304s | FedCP | $\Sigma + O(d^2)$ | 165s |
| **FedPAM** | $\Sigma$ | 143s | – | – | – |

**Runtime and communication analysis.** Table 8 reports the runtime and communication cost of different methods on CIFAR-100. Although FedAS communicates parameters at the same scale as FedAvg, it additionally requires transmitting a state parameter $S$, which introduces extra overhead and results in a much slower runtime of 304s. FedPer and FedRep transmit only a subset of the shared model parameters $\Sigma$ together with a small personalized portion of size $\alpha\Sigma$. While this fraction is smaller than the full model, the additional training stages for personalized heads lead to longer runtimes of 246s. Methods with quadratic overhead such as pFedFDA and FedCP incur communication complexity $\Sigma + O(d^2)$, where $d$ is the feature dimension of the backbone output, thus further increasing the communication burden. In contrast, FedPAM only requires transmitting the standard model parameters $\Sigma$ as in FedAvg, and achieves competitive efficiency with 143s, much faster than most personalized baselines while still offering higher accuracy. These results demonstrate that FedPAM provides a better trade-off between communication efficiency and personalization quality.

## 5 CONCLUSION

We presented FedPAM, a lightweight personalization approach for FL that calibrates the shared classifier via PAM and reinforces this calibration with a supervised contrastive alignment objective. Operating directly at the interface between the classifier and the features while keeping the standard FedAvg protocol unchanged, FedPAM delivers robust personalization under heterogeneous data and partial participation with no additional communication cost. Across diverse image benchmarks, partition schemes, client scales, and dropout regimes, FedPAM attains top or near top accuracy, with especially strong gains in highly heterogeneous settings. Ablations show that PAM and the personalized contrastive loss are complementary: each helps on its own, and together they yield the largest improvements. From a systems perspective, local training time remains competitive and the communication footprint is identical to FedAvg.Extensive experiments show that FedPAM achieves significant improvements over state of the art approaches on diverse image classification tasks.

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

APPENDIX

We provide additional materials omitted from the main paper.

- **Appendix A**: algorithmic procedure of FedPAM (see Section A).
- **Appendix B**: details of the convergence analysis (see Section B).

## A ALGORITHMIC PROCEDURE OF FEDPAM

**Algorithm.** We summarize the overall procedure of our proposed FedPAM framework in Algorithm 1. The server coordinates multiple clients by broadcasting the global model, collecting local updates, and aggregating them, while each client optimizes its local objective with the PAM-adjusted classifier. The pseudocode below outlines the detailed workflow of both the server and clients.

---
**Algorithm 1** FedPAM
---
1: **procedure** SERVER($T, r, \theta^{(0)}, \mathbf{W}^{(0)}$)
2:     **for** $t = 0, \ldots, T - 1$ **do**
3:         Sample clients $\mathcal{S}_t$ with ratio $r$; send $(\theta^{(t)}, \mathbf{W}^{(t)})$.
4:         **for** $k \in \mathcal{S}_t$ **in parallel do**
5:             Receive $(\theta_k^{(t+1)}, \mathbf{W}_k^{(t+1)})$
6:         **end for**
7:         Aggregate:$\theta^{(t+1)} \leftarrow \sum_{k \in \mathcal{S}_t} \frac{|\mathcal{D}_k|}{\sum_{j \in \mathcal{S}_t} |\mathcal{D}_j|} \theta_k^{(t+1)}, \mathbf{W}^{(t+1)} \leftarrow \sum_{k \in \mathcal{S}_t} \frac{|\mathcal{D}_k|}{\sum_{j \in \mathcal{S}_t} |\mathcal{D}_j|} \mathbf{W}_k^{(t+1)}$
8:     **end for**
9: **end procedure**
10: **procedure** CLIENT $k$(Receive $(\theta, \mathbf{W})$, data $\mathcal{D}_k$, local epochs $E$)
11:     **if** first participation **then**
12:         Initialize $\mathbf{P}_k \leftarrow \mathbf{I}_d$
13:     **end if**
14:     **for** $e = 1$ to $E$ **do**
15:         **for** minibatch $(\mathbf{x}, y) \subset \mathcal{D}_k$ **do**
16:             $\mathbf{z} \leftarrow f_\theta(\mathbf{x})$
17:             $\mathbf{W}_k^{\mathrm{adj}} \leftarrow \mathbf{W}\mathbf{P}_k$
18:             $\mathcal{L} \leftarrow \mathcal{L}_{\mathrm{ce}}(\mathbf{W}\mathbf{z}, y) + \lambda \mathcal{L}_{\mathrm{pcl}}(\mathbf{W}_k^{\mathrm{adj}}, \mathbf{z}, y)$
19:             Update $(\theta, \mathbf{W}, \mathbf{P}_k) \leftarrow (\theta, \mathbf{W}, \mathbf{P}_k) - \eta \nabla \mathcal{L}$
20:         **end for**
21:     **end for**
22: **end procedure**
---

## B DETAILS OF THE CONVERGENCE ANALYSIS

In this section, we present the complete convergence analysis for Algorithm 1 under the full participation assumption. For simplicity of notation, we use $\theta$ to denote $(\theta, W)$ throughout the analysis.

**FedAvg Update.** At the beginning of communication round $t$, the server broadcasts the global model $\theta_t$ to a subset of clients. Let $\mathcal{S}_t \subseteq [K]$ denote the set of selected clients, and let $\{p_k\}_{k=1}^K$ be the nonnegative weights with $\sum_{k=1}^K p_k = 1$ (often chosen proportional to local dataset sizes).

Within round $t$, every selected client $k \in \mathcal{S}_t$ initializes $\theta_{k,0} = \theta_t$ and performs $E$ steps of local SGD:

$$\theta_{k,\tau+1} = \theta_{k,\tau} - \gamma\, g_{k,\tau}, \qquad \tau = 0, \ldots, E - 1, \tag{B.1}$$

where $g_{k,\tau}$ is the stochastic gradient on client $k$ at local step $\tau$.

Afterwards, the server aggregates the updated local models to form the next global model:

$$\theta_{t+1} = \sum_{k \in \mathcal{S}_t} p_k\, \theta_{k,E}. \tag{B.2}$$

**Lemma 1** (One Local Step). Under Assumptions 1–2–3, for any local iterate, we have

$$\tilde{F}_k(\theta_{k,\tau+1}) \leq \tilde{F}_k(\theta_{k,\tau}) \;-\; \gamma\Big(1 - \tfrac{\tilde{L}\gamma}{2}\Big)\big\|\nabla\tilde{F}_k(\theta_{k,\tau})\big\|^2 \;+\; \tfrac{\tilde{L}\gamma^2\tilde{\sigma}^2}{2}. \tag{B.3}$$

*Proof.* Let the stochastic local gradient at step $\tau$ be

$$g_{k,\tau} \;=\; \nabla_\theta \ell_k^{\mathrm{CE}}(\theta_{k,\tau}) + \lambda\,\nabla_\theta \ell_k^{\mathrm{pcl}}(\theta_{k,\tau}).$$

Under Assumption 2, $\mathbb{E}[g_{k,\tau} \mid \theta_{k,\tau}] = \nabla\tilde{F}_k(\theta_{k,\tau})$. Under Assumption 3, we have the variance bound

$$\mathbb{E}\big[\|g_{k,\tau} - \nabla\tilde{F}_k(\theta_{k,\tau})\|^2\big] \;\leq\; \tilde{\sigma}^2.$$

By $\tilde{L}$-smoothness of $\tilde{F}_k$ (Assumption 1),

$$\tilde{F}_k(\theta_{k,\tau+1}) \leq \tilde{F}_k(\theta_{k,\tau}) + \big\langle \nabla\tilde{F}_k(\theta_{k,\tau}),\, \theta_{k,\tau+1} - \theta_{k,\tau}\big\rangle + \tfrac{\tilde{L}}{2}\|\theta_{k,\tau+1} - \theta_{k,\tau}\|^2.$$

Using the local update $\theta_{k,\tau+1} = \theta_{k,\tau} - \gamma\,g_{k,\tau}$ and by Assumption 2,

$$\mathbb{E}\Big[\tilde{F}_k(\theta_{k,\tau+1}) \,\Big|\, \theta_{k,\tau}\Big] \leq \tilde{F}_k(\theta_{k,\tau}) - \gamma\|\nabla\tilde{F}_k(\theta_{k,\tau})\|^2 + \tfrac{\tilde{L}\gamma^2}{2}\,\mathbb{E}\big[\|g_{k,\tau}\|^2 \,\big|\, \theta_{k,\tau}\big]. \tag{B.4}$$

By variance decomposition and Assumption 3,

$$\mathbb{E}\big[\|g_{k,\tau}\|^2 \,\big|\, \theta_{k,\tau}\big] \leq \|\nabla\tilde{F}_k(\theta_{k,\tau})\|^2 + \tilde{\sigma}^2. \tag{B.5}$$

From equation equation B.5, substituting it into equation B.4, we can obtain equation B.3. After rearrangement, we can obtain equation B.3. $\qquad\square$

**Lemma 2** (Telescoping over $E$ Local Steps). Then for $\tilde{\beta} := \gamma\big(1 - \tfrac{\tilde{L}\gamma}{2}\big)$ with $\gamma \in (0, \tfrac{2}{\tilde{L}})$, for each aggregation, we have

$$\mathbb{E}\,\tilde{F}(\theta_{t+1}) \;\leq\; \tilde{F}(\theta_t) - \tilde{\beta}\,\mathbb{E}\|\nabla\tilde{F}(\theta_t)\|^2 + \frac{\tilde{L}\gamma^2 E}{2}\,\tilde{\sigma}^2 + \frac{\tilde{L}^2\gamma^3}{12}\,E(E-1)(2E-1)\,\tilde{\sigma}^2. \tag{B.6}$$

*Proof.* From Lemma 1, for each client $k$ and local step $\tau$, summing over $\tau = 0, \ldots, E-1$ gives

$$\mathbb{E}\,\tilde{F}_k(\theta_{k,E}) - \tilde{F}_k(\theta_t) \leq -\gamma\Big(1 - \tfrac{\tilde{L}\gamma}{2}\Big)\sum_{\tau=0}^{E-1}\mathbb{E}\|\nabla\tilde{F}_k(\theta_{k,\tau})\|^2 + \tfrac{\tilde{L}\gamma^2 E\,\tilde{\sigma}^2}{2}.$$

Taking the weighted average across clients ($\tilde{F} = \sum_k p_k\tilde{F}_k$) yields

$$\sum_k p_k\mathbb{E}\,\tilde{F}_k(\theta_{k,E}) - \tilde{F}(\theta_t) \leq -\gamma\Big(1 - \tfrac{\tilde{L}\gamma}{2}\Big)\sum_{\tau=0}^{E-1}\sum_k p_k\,\mathbb{E}\|\nabla\tilde{F}_k(\theta_{k,\tau})\|^2 + \tfrac{\tilde{L}\gamma^2 E\,\tilde{\sigma}^2}{2}. \tag{B.7}$$

To proceed, we bound the double sum. For each $\tau$, note that the weighted sum of squares is always no smaller than the square of the weighted sum, i.e.,

$$\sum_k p_k\|\nabla\tilde{F}_k(\theta_{k,\tau})\|^2 \;\geq\; \Big\|\sum_k p_k\nabla\tilde{F}_k(\theta_{k,\tau})\Big\|^2.$$

We decompose

$$\sum_k p_k\nabla\tilde{F}_k(\theta_{k,\tau}) = \nabla\tilde{F}(\theta_t) + \sum_k p_k\big(\nabla\tilde{F}_k(\theta_{k,\tau}) - \nabla\tilde{F}_k(\theta_t)\big).$$

By $\tilde{L}$-smoothness,

$$\|\nabla\tilde{F}_k(\theta_{k,\tau}) - \nabla\tilde{F}_k(\theta_t)\| \leq \tilde{L}\|\theta_{k,\tau} - \theta_t\| = \tilde{L}\gamma\Big\|\sum_{s=0}^{\tau-1} g_{k,s}\Big\|.$$

Taking expectation and using the variance bound (assumption 3) on stochastic gradients yields

$$\mathbb{E}\|\nabla\tilde{F}_k(\theta_{k,\tau}) - \nabla\tilde{F}_k(\theta_t)\|^2 \;\leq\; \tilde{L}^2\gamma^2\tau^2\,\tilde{\sigma}^2.$$

Summing this over $\tau = 0, \ldots, E - 1$ and using $\sum_{\tau=0}^{E-1} \tau^2 = \frac{E(E-1)(2E-1)}{6}$, we obtain the drift penalty

$$\frac{\tilde{L}^2 \gamma^2}{12} E(E-1)(2E-1) \tilde{\sigma}^2.$$

Therefore,

$$\sum_{\tau=0}^{E-1} \sum_k p_k \, \mathbb{E} \|\nabla \tilde{F}_k(\theta_{k,\tau})\|^2 \geq \sum_{\tau=0}^{E-1} \left( \mathbb{E} \|\nabla \tilde{F}(\theta_t)\|^2 - \mathbb{E} \|\nabla \tilde{F}_k(\theta_{k,\tau}) - \nabla \tilde{F}_k(\theta_t)\|^2 \right)$$

$$\geq E \, \mathbb{E} \|\nabla \tilde{F}(\theta_t)\|^2 - \mathbb{E} \|\nabla \tilde{F}_k(\theta_{k,\tau}) - \nabla \tilde{F}_k(\theta_t)\|^2.$$

Combining the descent bound equation B.7, the drift penalty, and the averaging gap gives

$$\mathbb{E} \, \tilde{F}(\theta_{t+1}) - \tilde{F}(\theta_t) \leq -\tilde{\beta} \, \mathbb{E} \|\nabla \tilde{F}(\theta_t)\|^2 + \frac{\tilde{L}^2 \gamma^3}{12} E(E-1)(2E-1) \tilde{\sigma}^2 + \frac{\tilde{L} \gamma^2 E}{2} \tilde{\sigma}^2,$$

which proves the claim. $\qquad \square$

**Main Theorem.** Here we provide the proof of Theorem 1.

Suppose Assumptions 1–3 hold, $\tilde{F}$ is bounded below by $\tilde{F}_{\inf}$, then for any horizon $T \geq 1$,

$$\frac{1}{T} \sum_{t=0}^{T-1} \mathbb{E} \|\nabla \tilde{F}(\theta_t)\|^2 \leq \frac{\tilde{F}(\theta_0) - \tilde{F}_{\inf}}{\tilde{\beta} \, T} + \frac{\tilde{L} \gamma}{2\tilde{\beta}} E \tilde{\sigma}^2 + \frac{\tilde{L}^2 \gamma^2}{12 \tilde{\beta}} E(E-1)(2E-1) \tilde{\sigma}^2. \qquad \text{(B.8)}$$

*Proof.* From Lemma 2, summing this inequality over $t = 0, \ldots, T - 1$ and telescoping yields

$$\tilde{F}(\theta_T) - \tilde{F}(\theta_0) \leq -\tilde{\beta} \sum_{t=0}^{T-1} \mathbb{E} \|\nabla \tilde{F}(\theta_t)\|^2 + \left( \frac{\tilde{L} \gamma^2 E}{2} + \frac{\tilde{L}^2 \gamma^3}{12} E(E-1)(2E-1) \right) \tilde{\sigma}^2 \, T.$$

Since $\tilde{F}(\theta_T) \geq \tilde{F}_{\inf}$, we rearrange to obtain

$$\sum_{t=0}^{T-1} \mathbb{E} \|\nabla \tilde{F}(\theta_t)\|^2 \leq \frac{\tilde{F}(\theta_0) - \tilde{F}_{\inf}}{\tilde{\beta}} + \left( \frac{\tilde{L} \gamma^2 E}{2} + \frac{\tilde{L}^2 \gamma^3}{12} E(E-1)(2E-1) \right) \frac{\tilde{\sigma}^2 T}{\tilde{\beta}}.$$

Finally, dividing through by $T$ gives

$$\frac{1}{T} \sum_{t=0}^{T-1} \mathbb{E} \|\nabla \tilde{F}(\theta_t)\|^2 \leq \frac{\tilde{F}(\theta_0) - \tilde{F}_{\inf}}{\tilde{\beta} T} + \frac{\tilde{L} \gamma^2}{2\tilde{\beta}} E \tilde{\sigma}^2 + \frac{\tilde{L}^2 \gamma^3}{12 \tilde{\beta}} E(E-1)(2E-1) \tilde{\sigma}^2.$$

which is exactly equation B.8. $\qquad \square$

