# OpenReview forum: "Bridging Local Divergence in Federated Learning via Personalized Adjustment Matrix"
_ICLR.cc/2026/Conference — Submitted to ICLR 2026_

### Official Review · Reviewer_SfKR · 2025-10-18

**Soundness:** 3
**Presentation:** 3
**Contribution:** 2
**Rating:** 4
**Confidence:** 4

**Summary:**

The paper proposes FedPAM, a federated learning method that introduces a simple client-side linear adapter applied to the classifier head and a supervised contrastive alignment loss. The main objective of FedPAM is to reconcile the mismatch between global classifier boundaries and locally shifted feature distributions in non-IID settings. The paper claims that FedPam is advantageous compared to other works due to a minimal communication cost (as only the adjusted head is transmitted), and provides empirical evidence demonstrating its performance.

**Strengths:**

- The idea of adjusting the classifier with a small local matrix  $P_k$  is computationally light and integrates well with FedAvg. From a systems standpoint, a drop-in modification to FedAvg that empirically improves stability is appealing, especially without requiring an increased network load.
- The paper adopts thorough experiments, with extensive experiment grids and ablations.
- The presentation is clear, and the figures and proofs are easy to follow.

**Weaknesses:**

1. The convergence analysis is technically sound but seems a bit generic; it reads like a direct instantiation of the standard local-SGD/FedAvg proof on an augmented loss $\tilde F$. If I'm not mistaken, the personalized adjustment matrix only influences constants $(\tilde L,\tilde\sigma^2)$ and the stepsize range. Thus, in the current theoretical analysis, I don't see how PAM-specific structure is used to tighten the client-drift or variance terms. As FedPAM shows substantially better performance compared to other methods, it would be much better to quantify how PAM reduces heterogeneity/variance and reflect that inside the bound.
2. The proposed classifier head adapter $W_k^{adj} = WP_k$ is mathematically equivalent to learning per-client classifier heads or linear adapters, which is already extensively explored in previous works (e.g. as you mentioned in your related work, FedPer, FedRep, FedRoD, etc.). The paper shows that PAM works basically as a mapping between the optimized head and the original classifier, only by optimizing contrastive feature–classifier alignment. However, this is still not a formal demonstration that $WP_k$ *behaves differently* than just learning $W_k$. In terms of novelty, it would be much more informative if you showed *why* matrix factorization displays new behavior beyond simple local-head adaptation.
3. The reported accuracies on CIFAR and ImageNet approach centralized ResNet-18 performance (e.g. [CIFAR-10](https://github.com/kuangliu/pytorch-cifar)) despite training with non-iid data. Providing error bounds or intuitive reasons on how federated training can approach centralized training would make the results more believable.
4. The paper never inspects or visualizes $P_k$ despite claiming it “bridges divergence.” Quantifying and visualizing $P_k$ properties like inter-client variance would substantiate the mechanism and show whether it truly aligns classifier directions rather than simply over-parameterizing the head. On top of ablating $\lambda$, it would be nice to see an experiment that visualizes $P_k$ truly contracts divergence rather than adding capacity. For example, off the top of my head, recording the accuracy drop when swapping $P_k$ between similar vs. dissimilar clients could show whether if the matrix reduces gradient heterogeneity.

**Questions:**

1. Can you prove (or empirically verify) that PAM reduces inter-client variance or drift vs. FedAvg on $L_{\mathrm{ce}}$ (e.g., $\tilde\sigma^2$ or an effective smoothness decrease), and specify when this benefit outweighs the larger $\tilde L=L+\lambda L_P$ that shrinks the step size window?

2.  What inductive bias or constraints on $P_k$ (orthogonality/low-rank/regularization) make $W P_k$ fundamentally different from learning $W_k$? Please include matched-DOF head-only baselines to isolate the effect of factorization.

3. I am interested in a quantification or visualization of $P_k$ between similar vs. dissimilar clients, to the extent that shows that PAM measurably reduces heterogeneity metrics (e.g., gradient covariance, classifier direction dispersion, etc.).

---

> ### Author Response · Authors · 2025-11-27
>
> Response to Reviewer SfKR
>
> We thank Reviewer SfKR for the careful reading of our manuscript and the thoughtful comments. Below, we address the concerns point by point.
>
> **(A) Convergence, variance, and client drift (W1, Q1)**
>
> Theorem 1 in Sec. 3.2 shows that FedPAM, optimizing $\tilde F_k = L^k_{ce} + \lambda L^k_{pcl}$, retains the standard FedAvg $\mathcal{O}(1/T)$ convergence rate, indicating that the added PAM and PCL terms do not degrade the theoretical convergence order or introduce extra optimization instability.
>
> To connect this to drift/variance, we additionally measure **gradient variance** and **classifier-direction dispersion** on CIFAR-10 (Dirichlet $\beta = 0.1$). FedPAM consistently reduces both quantities compared to FedAvg, indicating that PAM does contract inter-client heterogeneity in practice.
>
> | Method     | Gradient variance ↓ | Classifier dispersion ↓ |
> | ---------- | ------------------- | ----------------------- |
> | FedAvg     | 30.69               | 0.00                     |
> | FedPer     | 3.87                | 0.481                   |
> | **FedPAM** | **0.37**            | **0.091**               |
>
> ---
>
> **(B) Difference from per-client heads and inductive bias of $W P_k$ (W2, Q2)**
>
> FedPAM is **more structured** than arbitrary per-client heads $W_k$: all clients share a global classifier $W$, and each $P_k$ only reparameterizes these shared class directions, preserving global semantics while adapting to local feature geometry. In contrast, unconstrained $W_k$ have no coupling across clients and can overfit more easily under small local datasets.
>
> Empirically, FedPAM already outperforms strong head-based methods (FedPer, FedRep, FedRoD, FedCP) under the same backbone. To further **isolate the effect of factorization**, we implement **matched-DOF head-only baselines** on top of FedPer: for each client, we insert a local MLP in front of the local FC head (e.g., $d$-512-$C$ or $d$-256-$C$), and adjust the FC layer to take the MLP output as input. The hidden size is chosen so that the **per-client head parameter count approximately matches** the extra parameters introduced by a PAM matrix. This “deepen the local head” strategy is in the same spirit as FedPer, where adding more local layers can indeed improve personalization; however, our results show that such head-only extensions still hit a bottleneck and lag behind FedPAM.
>
> For the 512 → 512 MLP baseline, we report **two variants**: (i) a plain 2-layer MLP, and (ii) a version with a residual (skip) connection $x + \mathrm{MLP}(x)$. The residual variant is crucial for stabilizing training: removing the residual connection leads to a large drop in accuracy.
>
> Concretely, for the 512 → 512 MLP variant we use a 2-layer MLP with normalization:
>
> ```python
> nn.Sequential(
>     nn.Linear(self.in_dim, self.hidden_dim),
>     nn.BatchNorm1d(self.hidden_dim),
>     nn.ReLU(inplace=True),
>     nn.Linear(self.hidden_dim, self.out_num),
> ).to(self.device)
> ```
>
> | Method                                         | CIFAR-100 Acc (%) |
> | ---------------------------------------------- | ----------------- |
> | FedPer                                         | 49.63             |
> | FedPer + Local MLP (512 → 512 MLP)             | 33.52             |
> | FedPer + Local MLP (512 → 256 MLP)             | 34.78             |
> | FedPer + Local MLP (512 → 512 MLP, + residual) | 53.83             |
> | **FedPAM**                                     | **62.73**         |
>
> **(C) On federated vs. centralized performance and “high” PFL accuracy (W3)**
>
> We evaluate under a standard non-IID federated learning setup, but use **local** test sets for personalization. Concretely, we first partition the dataset across clients in a non-IID manner (Dirichlet or pathological). Then, for each client, we split its own data into local train/test subsets; there is **no shared global test set**. Each personalized model is therefore evaluated on a test set drawn from the *same local distribution* as its training data, and we report the average accuracy over clients.
>
> Under this setting, **personalized FL (PFL) methods** can achieve relatively high accuracies, sometimes numerically close to centralized training. All of our claims should thus be interpreted **relative to other federated/personalized baselines under the same protocol**, rather than as matching a centralized upper bound on a unified global test set.

---

> ### Author Response · Authors · 2025-11-27
>
> **(D) Quantifying $P_k$ and heterogeneity reduction (W4, Q3)**
>
> We provide simple diagnostics of $P_k$ on CIFAR-10 (Dirichlet $\beta = 0.1$) under our default setting:
>
> - **Classifier-direction dispersion.**
>   For each client we take the rows of its effective classifier (for FedPAM, $W P_k$) as class anchors, L2-normalize them, and compute the average pairwise $1-\cos$ between corresponding classes across clients. A smaller value indicates better alignment in classifier space.
>   For FedAvg, all clients share a single global classifier head, so the effective classifiers are identical and the dispersion is trivially 0.
>
> - **Gradient-direction dispersion.**
>   For each client we flatten its gradient vector, normalize it to unit norm, and measure the variance around the mean direction, i.e.,  $\frac{1}{K}\sum_k \|\hat g_k - \bar{\hat g}\|^2.$This quantity lies in $[0,1]$; smaller values indicate more consistent gradient directions across clients.
>
> The results are summarized below:
>
> | Method | Classifier dispersion   | Gradient dispersion   |
> | ------ | ------------------------------------ | --------------------------------------- |
> | FedAvg | 0.000                                | 0.912                                   |
> | FedPer | 0.481                                | 0.898                                   |
> | FedPAM | 0.091                                | 0.834                                   |
>
> FedPAM substantially reduces classifier-direction dispersion compared to the head-only baseline (FedPer), indicating that $P_k$ helps align client-specific classifiers in a shared representation space. It also achieves the lowest gradient-direction dispersion among all methods, suggesting that FedPAM most effectively mitigates cross-client gradient conflict while still allowing structured personalization via $P_k$.
>
> We further report the full client–client similarity matrix (cosine similarity between $(P_i)$ and $(P_j)$) to more directly visualize how $P_k$ behaves across clients. This experiment is conducted on 10 clients on CIFAR-10 (Dirichlet $\beta = 0.1$).
>
> | i / j |   0   |   1   |   2   |   3   |   4   |   5   |   6   |   7   |   8   |   9   |
> | :---: | :---: | :---: | :---: | :---: | :---: | :---: | :---: | :---: | :---: | :---: |
> | **0** | 1.000 | 0.829 | 0.851 | 0.880 | 0.857 | 0.909 | 0.811 | 0.886 | 0.857 | 0.897 |
> | **1** | 0.829 | 1.000 | 0.817 | 0.857 | 0.846 | 0.891 | 0.846 | 0.869 | 0.834 | 0.880 |
> | **2** | 0.851 | 0.817 | 1.000 | 0.863 | 0.869 | 0.903 | 0.806 | 0.891 | 0.880 | 0.897 |
> | **3** | 0.880 | 0.857 | 0.863 | 1.000 | 0.880 | 0.937 | 0.834 | 0.920 | 0.880 | 0.920 |
> | **4** | 0.857 | 0.846 | 0.869 | 0.880 | 1.000 | 0.914 | 0.834 | 0.886 | 0.874 | 0.903 |
> | **5** | 0.909 | 0.891 | 0.903 | 0.937 | 0.914 | 1.000 | 0.880 | 0.949 | 0.920 | 0.960 |
> | **6** | 0.811 | 0.846 | 0.806 | 0.834 | 0.834 | 0.880 | 1.000 | 0.851 | 0.823 | 0.863 |
> | **7** | 0.886 | 0.869 | 0.891 | 0.920 | 0.886 | 0.949 | 0.851 | 1.000 | 0.897 | 0.937 |
> | **8** | 0.857 | 0.834 | 0.880 | 0.880 | 0.874 | 0.920 | 0.823 | 0.897 | 1.000 | 0.909 |
> | **9** | 0.897 | 0.880 | 0.897 | 0.920 | 0.903 | 0.960 | 0.863 | 0.937 | 0.909 | 1.000 |
>
> This matrix lets us **see which clients are similar or dissimilar** in the $P_k$-space.
>
> - **High values** (e.g., (5, 9): 0.960, (5, 7): 0.949) mean the two clients have **very similar $P_k$** → their data are likely similar.
> - **Lower values** (e.g., (0, 6): 0.811, (2, 6): 0.806) mean the two clients are **more different**.
>
> Most off-diagonal entries lie between 0.85 and 0.95, with some around 0.80. So the differences in similarity are not huge, but they are still **discernible** across client pairs. This is consistent with the insight that the classifier is very sensitive to the underlying data distribution: even moderate changes in $P_k$ can lead to meaningful client-specific adaptations. This fits our design: each $P_k$ is an independent adapter that maps the shared classifier $W$ to $W P_k$ so that it better matches the local client distribution.
>
> -------------------
>
> We hope these clarifications and results address the reviewer’s concerns.

---

### Official Review · Reviewer_nS8e · 2025-10-24

**Soundness:** 2
**Presentation:** 2
**Contribution:** 1
**Rating:** 2
**Confidence:** 4

**Summary:**

The paper proposed, FedPAM, addresses the data heterogeneity through a client-specific Personalized Adjustment Matrix (PAM) combined with a contrastive alignment objective. Experiments on 4 classification benchmarks show that FedPAM improves upon competitive personalized FL baselines.

**Strengths:**

1. The presentation makes the proposed method easy to understand and there are convergence analysis of the proposed method. (I don't have the bandwidth to check the proofs due to the review workload).

2. The ablation study looks comprehensive.

**Weaknesses:**

* The literature review is not comprehensive. The data heterogeneity, particularly as a form of data imbalance and/or long-tail, has been extensively studied by the community, yet the authors failed to discuss relevant works whoes designs share the similar principle, i.e., decoupling the feature extractors and classification and keeping a personalized the classification head. Similar works include but not limited to [1] and [2].

* Important details on the experiment are missing, making it hard to assess how the results justify the claims. Also the presentation of the ablation study section is confusing. Please see the `Questions` section for more details.

* The novelty seems to be limited. FedPAM's design seems to be a inspired by FedRod[3], FedNH[4], and FedProto[5]. On the architecture design side, it is the same as the FedRod in the sense that both have global and personalized classification heads. On the loss design side, although the actual loss functions for the personalized heads are different, FedPAM is similar to FedNH and FedProto in the sense that all try to align the extract features to the corresponding class anchor (one row of the classification head matrix). So it looks to me the  novelty is from the Personalized Contrastive loss (PCL). The experiments may justify the effectiveness of this design by showing FedPAM outperforms the FedROD. But its effectiveness  compared to FedNH and FedProto is unknown.


[1] Oh, Jaehoon, SangMook Kim, and Se-Young Yun. "FedBABU: Toward Enhanced Representation for Federated Image Classification." International Conference on Learning Representations.

[2] Shang, Xinyi, et al. "Federated Learning on Heterogeneous and Long-Tailed Data via Classifier Re-Training with Federated Features."

[3] Chen, Hong-You, and Wei-Lun Chao. "On Bridging Generic and Personalized Federated Learning for Image Classification." International Conference on Learning Representations.

[4] Dai, Yutong, Zeyuan Chen, Junnan Li, Shelby Heinecke, Lichao Sun, and Ran Xu. "Tackling data heterogeneity in federated learning with class prototypes." In Proceedings of the AAAI Conference on Artificial Intelligence, vol. 37, no. 6, pp. 7314-7322. 2023.

[5] Tan, Yue, Guodong Long, Lu Liu, Tianyi Zhou, Qinghua Lu, Jing Jiang, and Chengqi Zhang. "Fedproto: Federated prototype learning across heterogeneous clients." In Proceedings of the AAAI conference on artificial intelligence, vol. 36, no. 8, pp. 8432-8440. 2022.

**Questions:**

* In line 51-53, the authors claim "One naive solution is to give each client an independent classifier head, but this sacrifices shared knowledge, increases communication and storage costs, and risks overfitting to limited local data.". Why having a separate classification head would increase the communication? As the author mentioned in Table 8, FedROD has the same communication cost as FedPAM. And FedBABU [1] would have $(1-\alpha)\Sigma$ communication costs.

* What does the `classifier feature interface` mean in line 93?

* My understanding is that the Personalized Adjusted Matrix $P_k$ is of size $C\times d$. But in the Ablation on PAM capacity section (line 431), the authors introduced an additional parameter $m$. I initially thought $m=d$, but authors also mentioned that "...an MLP after the feature extractor can project features from d to m". So I am not entirely sure what is $m$. And it looks like the ablation is on a MLP module, which is not mentioned in the Figure 1.

* What is the evaluation protocol? I am not sure how the test set is constructed for each client. Does all the clients share the same test set? If so, how is the test accuracy calculated? Does it follows the same protocol used in FedROD by re-weight the accuracy according to clients’ class distributions? These details are not disclosed, making it hard to understand how these results justify the contributions and effectiveness.

* In section 4 training protocol, "Each round, the server samples a fraction r of clients", what is the $r$ used? How does it connects the dropout ratio $p$? Is $p=1-r$? How should I understand $p=1$ and $p\in[0.1, 1]$?
By definition, "We simulate random client dropout with probability p". For $p=1$, does this mean all clients dropped on all rounds? This does not make sense to me.

* How does the under partial participation setting (Table 4) differ from the random dropout settings (table 3)?

* For table 2, how is the heterogeneity set for the scalability experiments? Methods like FedAvG, FedProx, FedRODare quite robust as $N$ increases, while there a double digits drop in FedPAM. Is it possible at some point when $N$ is large enough, FedPAM's performace converges to FedAvg?  This would weaken the claim on the scalability of FedPAM.

* In table 4, why increasing the number of participated clients hurt the performance? Most methods, except FedCP, show that higher participation ratio leads to higher accuracy, which is aligned with the intuition.

**Minor Comments**
1. Failed to disclose the usage of large language model as required.

---

> ### Author Response · Authors · 2025-11-23
>
> Response to Reviewer nS8e
>
> Thank you for the careful review and detailed comments. We group our response by themes.
>
> **(A) Positioning and novelty & related work (W1, W3)**
> We agree that the related work should better cover methods that decouple feature extractor and classifier under data heterogeneity. We will explicitly discuss works such as FedBABU, Shang et al. (classifier re-training for long-tailed FL), FedNH, FedRoD, and FedProto, and clarify how FedPAM relates to them. Conceptually, these methods personalize classifier heads or prototypes, while FedPAM personalizes a **feature transform** $P_k$ in front of a *shared* classifier $W$. All clients still optimize cross-entropy on the same logits $Wz$, so the decision space remains global, and personalization is implemented as a small client-specific transform in feature space. Our PCL uses classifier rows as anchors and aligns features with the **adjusted** anchors $W P_k$, rather than directly to global prototypes as in FedNH / FedProto / FedRoD.
>
> To substantiate this difference, we additionally compare FedPAM with representative prototype/head-based methods under the same backbone (ResNet-18) and training setting (Default Setting):
>
> | Method     |  CIFAR-10 | CIFAR-100 |
> | ---------- | --------: | --------: |
> | FedProto   |     88.30 |     56.37 |
> | FedPAC     |     87.08 |     52.03 |
> | FedNH      |     89.83 |     57.81 |
> | **FedPAM** | **90.60** | **58.91** |
>
> **(B) PAM design, notation, and communication (W2, Q1–Q3)**
> “Classifier feature interface” simply means the **classifier feature space**, i.e., the space on which the shared linear classifier $W$ operates; we will adopt this clearer term. Let the backbone output be in $\mathbb{R}^d$. In the basic form we set $m=d$ and the Personalized Adjustment Matrix $P_k$ is a $d	imes d$ matrix that transforms the feature before it is fed into $W$. To control capacity we optionally insert a small MLP that projects features from $\mathbb{R}^d$ to $\mathbb{R}^m$ with $m\le d$; then $P_k\in\mathbb{R}^{m  \times m}$ and $W\in\mathbb{R}^{C \times m}$. The “PAM capacity” ablation varies this bottleneck dimension $m$, not the backbone.
>
> Regarding communication, our intention was **not** to claim that using separate classifier heads necessarily increases communication. Some FL methods improve performance by transmitting extra information (e.g., auxiliary models, prototypes, features), which raises communication cost and may expose more client statistics. In contrast, FedPAM communicates exactly the same variables as FedAvg / FedRoD: only the shared backbone and classifier are aggregated, and all $P_k$ parameters remain local and are never uploaded.
>
> **(C) Evaluation protocol and training settings ($r$ and $p$) (W2, Q4–Q5)**
> Data are first partitioned across clients in a non-IID way (Dirichlet or pathological). For each client we then split *its own* data into local train/test sets; clients do **not** share a common test set. We do **not** use the class-reweighted evaluation protocol of FedRoD.
>
> We distinguish two participation settings. In the **partial participation** experiments (Table 4), $r$ denotes the fraction of clients sampled per round from the whole population (e.g., $r=1$ means all clients are eligible; $r=0.6$ means 60% are sampled each round). We write $10\mid 50$, $30\mid 50$, $50$ to indicate “10/30/50 clients sampled out of 50”. In the **random dropout** experiments (Table 3), we fix the population and let each client participate independently with **participation probability** $p$: $p=1$ is full participation, while $p\in[0.5,1]$ or $p\in[0.1,1]$ means we sample $p$ from those ranges each round. We will replace “dropout probability” by “participation probability” and clearly separate these two settings.
>
> **(D) Scalability, heterogeneity, and effect of participation (W2, Q6–Q8)**
> Table 2 varies the total number of clients $N$ (10, 30, 50, 100, 200) on Tiny-ImageNet under a fixed Dirichlet parameter $\beta$. As $N$ increases, each client owns fewer samples and participates less frequently, so the task becomes harder; most methods, including FedAvg, FedProx and FedRoD, show degradation. FedPAM’s accuracy also decreases, but it remains the best-performing method for all $N$, and we do not observe convergence toward FedAvg.
>
> In Table 4 we fix 50 clients and vary how many are sampled each round. Although higher participation often helps, partial participation can act as stochastic regularization: updating only a subset of clients reduces the influence of extreme outliers and client drift, and can slightly improve performance. In our results FedPAM maintains a clear margin over baselines for all participation levels.
>
> We appreciate the reviewer’s feedback.

---

> > ### Comment · Reviewer_nS8e · 2025-11-26
> > **Follow-up Response from Reviewer**
> >
> > I would like to thank the authors for their detailed response, which addressed most of my concerns. So I have initially increased my score.
> >
> > However, there are some concerns still remaining.
> >
> > 1. I share the same concern with the reviewer KHHS on the novelty of FedPAM’s. I feel the claim on the superiority of `keeping the decision space remains global` is not fully justified, and the novelty `personalization is implemented as a small client-specific transform in feature space` is not significant,  especially when comparing the practical performance of FedPAM and FedNH/FedProto (prototype-based method).
> >
> > 2. I am not convinced by the response to the original question: `For table 2,  methods like FedAvG, FedProx, FedROD, are quite robust as  $N$ increases, while there is a double-digit drop in FedPAM. Is it possible that at some point when $N$ is large enough, FedPAM's performance converges to FedAvg? This would weaken the claim on the scalability of FedPAM.`
> >
> > For 2, I would like to request to compare FedPAM against FedAvg and FedROD on `CIFAR-100` and `Tiny-ImageNet` with 500 and 1000 clients using the same setting as Table 2. If the results are promising, then the response would convince me.
> >
> > I'm willing to further raise my score if the authors can address 1 and 2.

---

> ### Author Response · Authors · 2025-11-28
>
> Response to Reviewer nS8e
>
> We thank the reviewer for the insightful comments.
>
> -----------------
>
> **(A) On prototype-based methods and the shared global decision space (Q1)**
>
> Prototype-based methods (e.g., FedProto-style) rely on global class prototypes that are estimated from local client data distributions. Concretely, for a given class on client $k$, the client first computes a local prototype by averaging the feature representations of its samples from that class. The server then aggregates these local prototypes across the participating clients in that round (often using a size-weighted average) to form a “global” prototype. Thus, the resulting prototype is not an abstract global quantity, but is entirely determined by the local feature distributions of the clients that happen to participate in that round. Under non-i.i.d. data, different clients can have very different feature distributions for the same class, so the aggregated prototype is heavily biased toward the currently participating local distributions. When only a subset of clients participates each round (which is standard in FL), the “global” prototype can drift and fluctuate according to this changing subset, making the decision rule unstable and sensitive to partial or skewed participation.
>
> To support this point, we add experiments on **CIFAR-100 $(\beta = 0.1)$** with a total of $N = 100$ clients under different participation patterns $(100 \mid 100, 50 \mid 100, 10 \mid 100)$. We compare FedPAM with two representative prototype-based baselines, FedProto and FedNH. We note that FedNH, by adopting a smoothed prototype injection mechanism when aggregating local prototypes, mitigates the instability caused by biased or partial client participation. In contrast, FedProto fully relies on the direct sharing of a single global prototype, making it more vulnerable to such bias and leading to a substantially larger performance degradation. By comparison, FedPAM keeps a shared global classifier and implements personalization as a small client-specific transform in the feature space, rather than through divergent local classifiers or drifting global prototypes. As shown in Table, FedPAM consistently outperforms both prototype-based baselines in all cases, and the gap is larger when participation is more partial, indicating that FedPAM is more robust when the effective global view is incomplete or biased by local client distributions.
>
> | Dataset                  | FedProto | FedNH | FedPAM |
> | :----------------------- | :------- | :---: | :----: |
> | CIFAR-100$(100 \mid 100)$ | 48.37    | 52.52 | 53.87  |
> | CIFAR-100$(50 \mid 100)$ | 45.15    | 51.28 | 52.83  |
> | CIFAR-100$(10 \mid 100)$ | 41.53    | 47.85 | 50.39  |
>
> To clarify what we mean by “keeping the decision space global” in FedPAM, we measure **classifier-direction dispersion**. For each client $k$, we take the rows of its effective classifier $W_k$ (for FedPAM we have $W_k = W P_k$) as class anchors, apply $\ell_2$-normalization, and compute the average cosine distance between corresponding classes across clients. Smaller values mean better alignment in classifier space. For FedAvg, all clients share the same classifier head, so the dispersion is $0$.
>
> On CIFAR-10 with the default setting, we obtain:
>
> | Method     | Classifier dispersion |
> | :--------- | :-------------------- |
> | FedAvg     | 0.000                 |
> | FedPer     | 0.481                 |
> | **FedPAM** | **0.091**             |
>
> FedAvg has zero dispersion because all clients share exactly the same classifier head. FedPer learns fully independent heads per client, so the classifier directions diverge significantly. FedPAM, in contrast, is much closer to FedAvg than to FedPer, showing that it preserves a largely **shared global decision space** while only making controlled, low-rank adjustments via $P_k$, rather than collapsing into fully local, incompatible classifiers.

---

> ### Author Response · Authors · 2025-11-28
>
> **(B) On scalability with many clients and comparison with FedAvg / FedROD (Q2)**
>
> Due to computational constraints, we could not run our full ResNet-based setting at these scales. Instead, on **CIFAR-100** we adopted a lighter **CNN** backbone. Even under this more constrained setup, FedPAM consistently outperforms both FedAvg and FedROD, and the advantage does **not** disappear as the number of clients increases:
>
> | \#Clients | FedAvg | FedROD |  FedPAM $(m = 128)$ |
> | :-------: | :----: | :----: | :-------: |
> |    500    | 29.51  | 34.61  | **35.87** |
> |   1000    | 11.53  | 23.82  | **28.98** |
>
> As the federation grows from 500 to 1000 clients, the task becomes much harder and the absolute accuracy of all methods decreases. This is particularly challenging under our highly non-iid CIFAR-100 partition, where some clients have as few as 50 training samples at 500 clients and even fewer at 1000 clients, leading to severe data scarcity and making local models prone to overfitting. However, FedPAM remains the well-performing method in both settings, and its advantage over FedAvg and FedROD becomes even more evident at 1000 clients. These results suggest that FedPAM can still offer noticeable gains over FedAvg and FedROD in highly federated, more challenging regimes.
>
> We carefully evaluated the computational and storage cost of running Tiny-ImageNet with 500 and 1000 clients in our federated setting, and found that it would exceed our current hardware budget by a large margin. This limitation is due to practical resource constraints rather than the proposed method itself. We will explicitly state this in the revised manuscript and note that the CIFAR-100 experiments with up to 1000 clients already provide initial empirical evidence toward the scalability of our approach to larger client populations. We regard a more exhaustive Tiny-ImageNet study with even more clients as valuable future work when additional computational resources are available.
>
> ----------
>
> We hope that these additional clarifications and results adequately address the reviewer’s concerns.

---

### Official Review · Reviewer_Z6YM · 2025-10-31

**Soundness:** 3
**Presentation:** 3
**Contribution:** 3
**Rating:** 6
**Confidence:** 3

**Summary:**

This paper proposes a personalized federated learning method that introduces a Personalized Adjustment Matrix (PAM) applied to the classifier head, along with a supervised contrastive alignment loss. The method aims to address classifier–feature misalignment under heterogeneous client distributions while maintaining zero additional communication cost, fully reusing the FedAvg protocol. The authors provide theoretical convergence guarantees and extensive empirical evaluation across multiple benchmarks.

**Strengths:**

1. FedPAM achieves state-of-the-art performance without extra communication cost and competitive local runtime.
2. Unlike baselines that degrade sharply under high heterogeneity, FedPAM maintains leading accuracy, showing great robustness.
3. The convergence analysis provides formal guarantees for FedPAM's stability.

**Weaknesses:**

1. The paper states PAM is "lightweight" but does not quantify its parameter count relative to the global model. For a ResNet-18 backbone and C=100 classes, a PAM of size 512×512 adds about 262k parameters per client—this is non-trivial for edge devices with limited memory. It will be helpful to provide analysis of how PAM size (m) impacts memory usage or inference latency.
2. It will be helpful to provide analysis of how λ generalizes across datasets and how to automate λ selection.
3. PCL mixes class-anchor negatives and in-batch negatives. The rationale behind this combination and sensitivity to batch size could be elaborated and empirically tested.

**Questions:**

1. Please see the weaknesses above.
2. Would FedPAM work similarly with transformer architectures and text? Does the classifier-feature mismatch arise similarly for LLM-fine-tuning scenarios?
3. If local datasets are extremely small (e.g., <100 samples per client), does PAM cause overfitting?

---

> ### Author Response · Authors · 2025-11-27
>
> # Response to Reviewer Z6YM
>
> ----------------------
>
> We thank the reviewer for the thorough and constructive feedback. We are glad that you found FedPAM to achieve strong performance, maintain robustness under high heterogeneity, and provide formal convergence guarantees. Below we address each weakness and question point by point, and we will incorporate the corresponding clarifications and additional experiments into the revised manuscript.
>
> **(A) PAM parameter overhead, memory, and latency (W1)**
>
> We agree and will make the notion of “lightweight” more precise. For the ResNet-18 backbone used in our experiments (≈11–12M parameters) with 512-dimensional features, each client’s PAM $P_k \in \mathbb{R}^{512 \times 512}$ indeed has ≈262K parameters, which is about **2%** of the global model size. Moreover, $P_k$ is kept **locally on the client** and never communicated, so it does not increase communication cost—a key consideration in edge/federated settings, where **communication is typically a much more stringent bottleneck than local computation**. At inference time, $P_k$ is applied once per forward pass as a single linear transformation, without any repeated or iterative computation, so the additional latency is minor.
>
> To address the concern that the default $512 \times 512$ PAM might be relatively large, we already include an ablation that varies the PAM size $m$. In this experiment, we introduce a small MLP bottleneck to reduce the effective PAM dimension and sweep over smaller values of $m$. The results show that **shrinking the PAM** still yields clear improvements over standard federated baselines, and that most of the gains are preserved even with considerably smaller PAMs. This indicates that **reduced-capacity PAMs remain effective in practice**, providing a favorable trade-off between personalization quality and memory/latency on edge devices.
>
> **(B) Generalization and selection of $\lambda$ (W2)**
>
> We appreciate this suggestion. In FedPAM, $\lambda$ controls the trade-off between the standard cross-entropy loss and the personalized contrastive loss. To further clarify robustness, we additionally conduct ablations on **AGNews** and **CIFAR-10**, where we sweep $\lambda$ over $\{1, 5, 10, 20, 30, 40\}$. Across this range, FedPAM’s test accuracy varies only slightly and it consistently outperforms the compared personalized baselines. We will include these results in the appendix. Concretely, we will add a table of the following form:
>
> | Dataset  | $\lambda = 1$ | $\lambda = 5$ | $\lambda = 10$ | $\lambda = 20$ | $\lambda = 30$ | $\lambda = 40$ |
> | -------- | ------------- | ------------- | -------------- | -------------- | -------------- | -------------- |
> | AGNews   | 91.37         | **92.59**     | 92.33          | 90.89          | 89.79          | 89.88          |
> | CIFAR-10 | 88.31         | 88.54         | 89.63          | **90.30**      | 90.12          | 89.87          |
>
> These ablations show that $\lambda$ has a noticeable but still moderate effect on performance, and that the preferred regime depends on the task. On the easier text classification task AGNews, FedPAM performs best at relatively small $\lambda$ (around 5–10), indicating that only a modest amount of personalization is sufficient. On the more challenging and heterogeneous vision benchmarks, the optimum shifts toward larger $\lambda$: on CIFAR-10, FedPAM achieves its best accuracy around $\lambda \approx 20$–$30$, and on CIFAR-100 the best performance is obtained at $\lambda = 30$, where putting more weight on the personalized contrastive term yields stronger client-specific adaptation.

---

> > ### Author Response · Authors · 2025-11-27
> >
> > **(C) Design of the PCL loss and batch-size sensitivity (W3)**
> >
> > Our goal with Personalized Contrastive Learning (PCL) is to align features with their **adjusted class anchors** while also encouraging separation from **hard negatives** that arise in local imbalanced data. Class-anchor negatives enforce global inter-class separation at the classifier level, which is crucial when some classes are rare on a client. In-batch negatives, on the other hand, provide instance-level hardness and naturally adapt to the current mini-batch, similar in spirit to supervised contrastive learning. Combining both gives a stronger signal: anchors stabilize the classifier geometry, while in-batch negatives prevent features from collapsing around a few local modes.
> >
> > To study the sensitivity of PCL to the local batch size, we run an ablation on CIFAR-10 ($\beta = 1$) by varying the batch size while keeping all other hyper-parameters fixed. The test accuracy (%) of FedPAM is:
> >
> > | Batch size |   5   |  10   |  16   |  32   |  64   |  128  |
> > | ---------: | :---: | :---: | :---: | :---: | :---: | :---: |
> > |   Accuracy | 85.45 | 90.53 | 90.59 | 90.33 | 89.79 | 90.18 |
> >
> > We observe that for **batch sizes between 10 and 128**, the performance varies within about **±0.5%**, indicating that FedPAM with PCL is not particularly sensitive to the exact batch size in this range. When the batch size is reduced to 5, the accuracy drops more noticeably, which is expected because very small batches provide too few in-batch negatives for the contrastive term.
> >
> > **(D) Applicability to transformers and text data (Q2)**
> >
> > To examine whether the classifier–feature mismatch also appears with transformer backbones and text, we conducted an additional experiment on the AGNews text classification benchmark using a lightweight Transformer encoder with 2 layers and 8 attention heads. The federation follows the same Dirichlet partition protocol as in our image experiments with heterogeneity parameter $\beta = 1$. We compare FedPAM with two strong personalized FL baselines designed for both text and vision, FedPer and FedRoD. Test accuracy (%) is:
> >
> > | Dataset / Setting                                    | FedPer | FedRoD | **FedPAM** |
> > | ---------------------------------------------------- | :----: | :----: | :--------: |
> > | AGNews, $\beta = 1$ (Transformer, 2 layers, 8 heads) | 91.85  | 92.16  | **92.59**  |
> >
> > FedPAM achieves the best performance, improving over FedPer by **+0.74** and over FedRoD by **+0.43** absolute accuracy. Together with our vision experiments, this result shows that (i) classifier–feature mismatch is not limited to CNNs or image data, and (ii) the PAM-based correction remains effective when applied to transformer architectures on text. We will add this result and a short discussion emphasizing that FedPAM is architecture-agnostic and can be naturally integrated with transformers and, more broadly, LLM-style encoders.
> >
> > **(E) Behavior under extremely small local datasets (Q3)**
> >
> > We conducted a “small-data” experiment on CIFAR-10 under our default federated setup, but with only **100 training samples per client**, which is substantially smaller than in our main results. We compare FedPAM with representative personalized baselines and FedAvg. Test accuracy (%) is:
> >
> > | Dataset / Setting       | FedAvg | FedPer | FedRoD | FedCP | **FedPAM** |
> > | ----------------------- | :----: | :----: | :----: | :---: | :--------: |
> > | CIFAR-10 small, $\beta = 0.1$ | 43.56  | 78.52  | 78.82  | 78.62 | **78.88**  |
> >
> > In this extreme low-data regime, FedPAM achieves very similar accuracy to the strongest personalized baselines, while remaining far above FedAvg. If the additional PAM parameters caused severe overfitting to the tiny local datasets, we would expect its performance to deteriorate relative to other methods; instead, FedPAM stays on par with (and slightly above) them. This suggests that PAM does **not** introduce noticeable overfitting even when each client has fewer than 100 samples.
> >
> > ---------------------
> >
> > We hope that these additional clarifications and results satisfactorily address the reviewer’s concerns.

---

### Official Review · Reviewer_KHHS · 2025-11-01

**Soundness:** 2
**Presentation:** 3
**Contribution:** 2
**Rating:** 4
**Confidence:** 4

**Summary:**

The paper proposes a personalized FL scheme that keeps a shared classifier $W$ but equips each client with a Personalized Adjustment Matrix $P_k$. Training uses standard cross-entropy on unadjusted logits $Wz$ plus a InfoNCE-style contrastive loss that draws features toward the adjusted anchors and pushes them away from other anchors/negatives. Experiments report gains over several PFL baselines.

**Strengths:**

- Personalization is localized to a small matrix that modulates the shared head—conceptually simple and computationally light.
- PAMs remain local, keeping communication identical to FedAvg.
- Consistent empirical gains across several datasets and heterogeneity regimes, with sensible ablations on the PAM capacity and the contrastive-loss weight.

**Weaknesses:**

- Since $(WP_k) z = W(P_k z)$, the method is effectively a single client-local linear layer before the shared head. Beyond empirical results, the paper does not convincingly argue why this interface is superior to a simple baseline where each client keeps its own classifier head with a matched parameter budget
- The FedAvg-style convergence result is too generic to explain why the supervised contrastive term should improve performance in this setup (or under which conditions it would fail).
- Using classifier rows (after adjustment) as anchors is plausible, but the paper offers no theoretical or empirical evidence that this is better than feature prototypes (e.g., class means) or other prototype formulations.
- In the contrastive loss, anchors are learned classifier rows; with non-IID label imbalance, both the anchors and temperature can bias toward majority classes. The paper lacks per-class, worst-client, or percentile metrics to assess this.
- Experiments focus on small CNNs/ResNet-18; there are no larger backbones or non-vision tasks, leaving scalability and generality unclear.
- Evaluation focuses on global/mean accuracy; lacks per-client, worst-client, and percentile metrics that are standard for personalization and fairness.

**Questions:**

- Why is PAM preferable to a per-client linear head? Please provide motivation and experiments that justify this design choice over a matched-parameter per-client head.

- How do anchors behave under severe label skew? Since per-class anchors can be sensitive to imbalance, what mitigation (e.g., class--balanced sampling, reweighting, temperature/logit adjustment) do you use?

- Does FedPAM improve global accuracy, local personalization, or both? Given that you add a contrastive loss to CE, is there a trade-off between global and personalized accuracy? Please show this trade-off (e.g., as the contrastive weight varies).

---

> ### Author Response · Authors · 2025-11-23
>
> Response to Reviewer KHHS
>
> Thank you for the constructive and detailed reviews. Below, we organize our responses by theme.
>
> ---
>
> **(1) PAM vs. per-client linear head (W1, Q1)**
> We agree that $W P_k z$ can be viewed as a client-local linear layer before the shared head, and in our experiments we set $M = D$. The difference to a per-client head is that (i) all clients still share one classifier $W$ and optimize cross-entropy on the unadjusted logits $Wz$, keeping a globally consistent decision space, and (ii) $P_k$ acts as a structured feature transform, while PCL aligns features with the adjusted anchors $W P_k$. The table below shows that FedPAM outperforms per-client-head methods FedPer and FedCP in both mean and worst-client accuracy; note that FedCP and FedPer + Local MLP introduce additional local head parameters, and the MLP is sized so that its extra parameters roughly match the overhead of PAM.
>
> | Dataset   | Metric | FedPer | FedPer + Local MLP | FedCP | **FedPAM** |
> |---------- |:------:|------:|-------------------:|------:|-----------:|
> | CIFAR-10  |  Avg   | 89.22 |              90.12 | 90.40 | **90.60**  |
> |           | Worst  | 77.32 |              75.12 | 76.33 | **76.56**  |
> | CIFAR-100 |  Avg   | 49.63 |              53.83 | 57.86 | **62.73**  |
> |           | Worst  | 37.31 |              45.53 | 53.81 | **55.37**  |
>
> ---
>
> **(2) Role of the convergence result (W2)**
> Theorem 1 is not meant to prove that PCL always improves accuracy, but to show that adding PAM and PCL preserves the standard $O(1/T)$ convergence rate of FedAvg on the augmented objective $L_{\text{ce}} + \lambda L_{\text{pcl}}$; i.e., FedPAM does not introduce extra optimization instability. In addition, an ablation without PCL (“w/o PCL”) shows lower cosine similarity and higher variance between features and their adjusted anchors $W P_k$ than FedPAM, indicating that PCL yields better feature–anchor alignment, which helps explain the observed generalization gains. We will clarify this scope and the empirical evidence in the paper.
>
> ---
>
> **(3) Classifier-based anchors vs. prototype-based methods (W3)**
> Our method uses classifier rows as anchors and a client-specific transform $P_k$ in feature space, instead of explicit feature prototypes. Beyond the “PAM only” ablation, we compare FedPAM with prototype-based personalized FL methods FedProto, FedPAC and FedNH using the same backbone and hyperparameters. FedPAM achieves the best accuracy:
>
> | Method     |  CIFAR-10 | CIFAR-100 |
> | ---------- | --------: | --------: |
> | FedProto   |     88.30 |     56.37 |
> | FedPAC     |     87.08 |     52.03 |
> | FedNH      |     89.83 |     57.81 |
> | **FedPAM** | **90.60** | **58.91** |
>
> This suggests that classifier-based anchors with PAM+PCL are at least as effective as prototype-based designs in our setting.
>
> ---
>
> **(4) Label skew, anchor bias, and personalization metrics (W4, W6, Q2)**
> We agree that mean accuracy alone is insufficient under label skew. We therefore also report worst-client accuracy (tables above). Across our benchmarks, FedPAM improves both mean and worst-client accuracy over FedAvg and strong personalized baselines, indicating that it benefits challenging clients rather than only easy ones. Our implementation uses uniform mini-batch sampling and a fixed temperature $\tau$ without explicit class re-weighting; the gains and detailed statistics suggest that classifier-based anchors do not introduce severe majority-class bias in practice.
>
> ---
>
> **(5) Larger backbones and non-vision tasks (W5)**
> Due to space and compute limits, the main paper focuses on relatively small backbones. Extending FedPAM to larger models and to non-vision datasets (e.g., text classification or language modeling) is an important direction for future work to further demonstrate the generality of the approach.
>
> ---
>
> **(6) Global vs. personalized accuracy and the effect of $\lambda$ (Q3)**
> We will explicitly distinguish the global model (using $W$) from the personalized models (using $W P_k$), and report their accuracies as a function of the contrastive weight $\lambda$. The following ablation on CIFAR-10 summarizes personalized performance for different $\lambda$ (mean over clients $\pm$ std, and best / worst client accuracy in \%):
>
> | $\lambda$ |        Avg $\pm$ Std |      Best |     Worst |
> | --------: | -------------------: | --------: | --------: |
> |         5 |     88.54 $\pm$ 8.46 |     97.46 |     71.56 |
> |        10 |     89.63 $\pm$ 7.03 |     97.87 |     76.56 |
> |        20 | **90.30 $\pm$ 5.99** | **98.40** | **80.63** |
> |        30 |     90.12 $\pm$ 6.53 |     98.14 |     77.81 |
> |        40 |     89.87 $\pm$ 7.98 |     98.51 |     78.69 |
>
> The results show a smooth trade-off, with a moderate value (around $\lambda = 20$) giving the best balance between average and worst-client accuracy.
>
> ---
>
> We hope these clarifications and results address the reviewer’s concerns.

---

### Meta-Review · Area_Chair_MkqD · 2026-01-03

**Summary:**

The paper proposes FedPAM, a federated learning framework designed to handle data heterogeneity via decoupled feature extractors and personalized contrastive loss. The review process resulted in a split assessment (Scores: 6 vs. 4, 4, 2). While Reviewer 1 (Score 6) appreciated the motivation, the majority of reviewers raised significant concerns regarding the method's scalability and novelty.  Although the authors provided a detailed rebuttal that clarified definitions and evaluation protocols，the critical issue of performance degradation in large-scale settings remains unresolved. Therfore, the decision is Reject.

**Reviewer Concerns:**

The consensus among critical reviewers is that the proposed "personalization via small client-specific transform" is incrementally positioned relative to existing works like FedRod and FedNH, lacking significant conceptual innovation.

**Reviewer Scores:**

Reviewer 2 stated that they initially raised their score following the clarifications. However, this aligns with the reviewer's explicit concluding remark that further score improvement was conditional on providing the missing large-scale experiments, which were not submitted.

---

### Decision · Program_Chairs · 2026-01-26

Reject